# Effect of Milk-Based Infant Formula Fortified with PUFAs on Lipid Profile, Growth and Micronutrient Status of Young Children: A Randomized Double-Blind Clinical Trial

**DOI:** 10.3390/nu13010004

**Published:** 2020-12-22

**Authors:** Marta Rivera-Pasquel, Mario Flores-Aldana, María-Socorro Parra-Cabrera, Amado David Quezada-Sánchez, Armando García-Guerra, Jorge Maldonado-Hernández

**Affiliations:** 1Centro de Investigación en Nutrición y Salud, Instituto Nacional de Salud Pública, Cuernavaca 62100, Mexico; mrivera@insp.mx (M.R.-P.); garciaf@insp.mx (A.G.-G.); 2Centro de Investigación en Salud Poblacional, Instituto Nacional de Salud Pública, Cuernavaca 62100, Mexico; mparra@insp.mx; 3Centro de Investigación en Evaluación y Encuestas, Instituto Nacional de Salud Pública, Cuernavaca 62100, Mexico; amado.quezada@insp.mx; 4Unidad de Investigación Médica en Nutrición, Hospital de Pediatría, Centro Médico Nacional Siglo XXI, IMSS, Ciudad de México 06720, Mexico; jormh@yahoo.com.mx

**Keywords:** DHA, PUFAs, lipids, infant formula, micronutrients

## Abstract

Background: Polyunsaturated fatty acids (PUFAs) are essential to child growth and development. Objective: To assess the effect of PUFAs-fortified infant formula on lipid profile, growth and micronutrient status in children 12 to 30 months old. Methods: This study is a double-blind randomized controlled clinical trial. Two study groups were assessed: (a) milk-based infant formula with micronutrients and PUFAs (PUFAs) and (b) milk-based infant formula with micronutrients, no PUFAs added (Non-PUFAs). Children received prepared formula (240 mL) twice a day, according to the color-code assigned to each infant. Anthropometric measurements and venous blood samples were taken at each day-care center at baseline, and again after four months. Total serum lipid extraction was 0.5 mL. Samples were treated and modified by the Folch method and analyzed with gas chromatography. Results: Changes in serum lipid profile (expressed as % FA) between baseline and four months showed a statistically significant increase in docosahexaenoic acid (DHA) (0.22 vs. −0.07, *p* < 0.05) and Alpha-Linoleic acid (0.08 vs. 0.02, *p* < 0.05) in infants who consumed PUFAs-fortified formula compared to Non-PUFAs-fortified formula. Infants increased their length/height-for-age Z-score: median change for the PUFAs group was 0.16 (95% CI = 0.08, 0.28) and 0.23 (95% CI = 0.14, 0.33) for Non-PUFAs, with no differences between groups. Median folate level was significantly higher among the PUFAs group compared to Non-PUFAs: −0.87 (95% CI = −1.38, −0.44) and −3.83 (95% CI = −4.65, −3.03) respectively. Consumption of both supplements was adequate and stable during the intervention. Conclusion: A significant improvement was observed in the lipid profile of children who received the PUFAs-fortified milk-based formula.

## 1. Introduction

Dietary lipids, especially long-chain polyunsaturated essential fatty acids (LCPUFAs), have received attention in the last several decades due to their influence on health [1]. During pregnancy and lactation, N-3 fatty acid (α-linolenic acid (ALA), eicosapentaenoic acid (EPA)) and docosahexaenoic acid (DHA) have been linked to length of gestation, preterm birth, birth weight, and postpartum depression [2]. During infancy and childhood, they have been associated with postnatal growth, cognitive and visual development, and allergies [3]. These essential fatty acids can be found mainly in oil-rich fish, breast milk, and algae [4]. The amount of essential fatty acids in human milk is determined mainly by the regular consumption of oil-rich fish, algae, and supplements during pregnancy and lactation [5].

The fatty acid profile can be influenced by maternal dietary habits and body stores [6]. A study carried out in pregnant Mexican women showed low intakes of preformed DHA, where median intake was 55 mg/day [6]. It is known that in pregnant and lactating women circulation levels of polyunsaturated essential fatty acids (PUFAs) are transferred to the fetus and infant through the placenta and/or mammary glands. Low levels of PUFAs may compromise maternal stores [7]. According to the National Health and Nutritional Survey-2006 in Mexico, more than 50% of the population (all ages) does not meet the recommended dietary intake of PUFAs. Median consumption of DHA and EPA were 30 mg/day [8].

Infants and children also suffer from micronutrient deficiencies, which have adverse effects on survival, growth, and development. The principal comorbidities include congenital defects, blindness, and greater susceptibility to infections, cognitive impairment, and premature mortality [9]. The most common micronutrient deficiencies worldwide among infants and young children are those related to vitamin A, folates, zinc, and fatty acids [9]. The estimated global prevalence of vitamin A deficiency in children <5 years of age is about 30% [10], while the prevalence of zinc deficiency in pre-school-age children is 17.4% [9]. Recent studies have shown vitamin D deficiency (25-OH-D < 50 nmol/L) in 33% of pre-school and school-age children [11]. In Mexico, the most prevalent nutrient deficiencies in children <5 years old include: iron (26%), zinc (28%) and anemia (20.6%), followed by moderate deficiencies in vitamin B12 (7.3%) and folic acid (3.6%) [12].

During infancy, it is difficult to reach the requirements of essential fatty acids and certain micronutrients. Many interventions have been implemented to improve LCPUFA and micronutrient status in children [9,13]. Supplementation and food fortification have been used to increase fatty acid and micronutrient consumption [9]. Many food vehicles include cereals, porridges, micronutrients powders, lipid-based nutrient supplements, and fortified milk [13].

A recent study in Mexico showed a decrease in the prevalence of anemia and iron deficiency in children 12 to 30 months who received iron-fortified milk as part of the national program Leche Industrializada Conasupo S.A. (LICONSA) [14]. This study led to the scaling up of a subsidized fortified milk distribution program, which benefits 4.2 million children from ages one to 11 years in Mexico [14]. Milk is an efficient vehicle because it is widely accepted by children and easy to prepare [14].

The main objective of this study is to assess the effect of PUFAs-fortified milk-based infant formula on lipid levels in Mexican infants 12 to 30 months old. Moreover, anthropometry and micronutrient status of the infants were evaluated.

## 2. Materials and Methods

### 2.1. Study Design and Intervention

This is a double-blind, randomized controlled trial with two treatment groups: (a) milk-based infant formula with micronutrients and PUFAs (PUFAs) and (b) milk-based infant formula with micronutrients only, and no PUFAs added (Non-PUFAs) (Table 1). The intervention period lasted for four months. The study was conducted in 2012–2013 in day-care centers in Cuernavaca, Morelos. The study was approved by the Ethics, Research, and Biosecurity Committees of the National Institute of Public Health in Mexico (in Spanish, INSP) (CONBIOÉTICA: 17CEI00120130424). Written informed consent for study participation was obtained from subjects’ parent or caregiver. Formula was provided free of charge to the parents/caregivers even if they dropped out of the study or did not want to participate. Parents/caregivers did not receive compensation for participation. Parents/caregivers received information on their child’s health and nutritional assessment. Recommendations on an adequate diet were provided at the end of the study. Parents/caregivers read the informed consent. If parents/caregivers accepted to participate in the study, they and a witness signed the informed consent. A copy of the signed informed consent was provided to parents/caregivers. Trial registration was obtained by Clinical Trials U.S. National Institutes of Health, NCT03397485 on 14 September 2017.

### 2.2. Population and Setting

Infants were recruited at 12 day-care centers in Cuernavaca, Morelos, México. Eligible infants were 12 to 30 months old, healthy, and whose parents/caregivers consented to study participation. The day-care centers were part of a national government program whose main objective was to support working mothers and single parents. Exclusion criteria were infants who were breastfed at the time of the study, infants receiving other milk with micronutrients, and those who were clinically sick. A hemoglobin screening test was conducted, and infants with capillary hemoglobin concentration <9.0 g/dL were excluded and referred for treatment at a local health center.

The study protocol was explained to the day-care directors, who agreed to participate. Each director made appointments with parents/caregivers at each day-care center. The project coordinator and a scientist-researcher explained the objective and methods of the study. At these appointments, parents/caregivers were invited to participate.

### 2.3. Randomization and Masking

The Moses Oakford method [15] was used to randomly assign each infant to one of the two groups to receive either milk-based infant formula with PUFAs (PUFAs), or a formula with micronutrients only (Non-PUFAs). The cans of formula were numbered consecutively to follow the random assignment. Both infant formulas had the same color (white milk powder), odor, and flavor, and were indistinguishable except for the color-coding of the can. A four-color code was used, two for each treatment: red & gray for the PUFAs treatment, and blue & green for the Non-PUFAs treatment. The color code was unknown to researchers, field workers, users, and analysts until the study ended. The manufacturer logo was not identifiable on the cans.

### 2.4. Product Preparation and Volume Intake

Milk-based infant formula was prepared Monday through Friday at each day-care center by trained personnel according to World Health Organization (WHO) guidelines. Formulas were reconstituted with 240 mL of purified water and eight spoonfuls of milk powder (5 g each spoonful; 40 g total powder). Formula was prepared in the bottles, then weighed using a measuring scale. Both groups received 480 mL per day: 240 mL at 7:00 a.m. and 240 mL at 4:00 p.m. Leftover formula was weighed using a measuring scale and recorded on a consumption form by the study personnel.

On weekends, parents/caregivers were instructed on how to prepare powdered milk infant formula according to WHO guidelines [16]. Study personnel provided them with the amount of infant formula necessary to prepare 480 mL per day over the weekend. They were trained to estimate and record infant formula consumption on a form with a sketch of an 8-ounce bottle marked for volume. This was also the case for infants who drank formula before arriving to the day-care center early in the morning. On Mondays, parents/caregivers handed in the consumption form to the project supervisor.

### 2.5. Outcomes

The main objective of the study was to assess the effect of milk-based infant formula fortified with PUFAs and micronutrients on lipid status between study enrollment and four months, as compared to milk-based formula with micronutrients only. Lipid profile included fatty acids with a chain length between 12 and 22 carbon atoms. Growth and micronutrient status were also assessed.

### 2.6. Measurement of Blood Samples

At the day-care centers, blood samples were obtained by two trained nurses according to protocol procedures established by the Biosecurity Commission at the National Institute of Public Health in Mexico (INSP).

Hemoglobin concentration was taken only at baseline for screening procedures, and was determined in capillary blood samples obtained by finger prick and measured in a Portable Photometer—Hemocue [17]. Cutoff points for anemia were defined according to WHO standards [18] and adjusted by altitude [19]. A 10 mL venous blood sample was also obtained from the antecubital vein at baseline, and at 4 months thereafter. Venous samples were centrifuged, using a portable centrifuge EBA8 (Hettich, Tuttlingen, Germany) at 280× *g*; 20 min in situ and serum was separated and stored in color-coded cryovials, preserved in liquid nitrogen until delivery to a central laboratory in Cuernavaca, Mexico and stored at −70 °C until defrosted for analyses.

### 2.7. Fatty Acid Profile

For extraction of total lipid content, 0.5 mL of plasma was treated with the modified Folch method [20]. Analysis was performed with a gas chromatograph 7820A (Agilent Technologies, Santa Clara, CA, USA) with a flame ionization detector (FID). Fatty acids were separated using a HP-88 capillary column (100 m × 0.25 mm ID; Agilent Technologies, Inc., USA). PUFAs ranging from 12–22 chain length carbon atoms and peaks were identified by comparing their retention times with those of high purity (>99%) standard mixtures (Sigma-Aldrich Chemie GmbH, 37 FAs mixture). Results were expressed as a percentage of all fatty acids detected with a chain length between 12 and 22 carbon atoms [21,22].

### 2.8. Micronutrients

Serum concentrations of ferritin (µg/L) were measured by an immunoassay method using commercial kits (Dade Behring Inc., Deerfield, IL, USA), while concentrations of zinc (µg/dL) were determined by atomic absorption spectrometry using an Analyst 300 spectrometer (Perkin-Elmer, Norwalk, Ct, USA) [23]. Folate (ng/mL) was transformed into cyanocobalamin and stable folates. Concentrations were measured by Toyo Soda Manufacturing Company (TOSOH) automated immunoanalyzer [24]. Vitamin D measured through 25-OH-D3 (nmol/L) was analyzed by chemiluminescent microparticle immunoassay (CMIA) using an Abbott Architect module [25,26]. Vitamin A (µg/dL) determinations were performed by high-performance liquid chromatography (HPLC) in a Waters instrument (Waters Co., Milford, MA, USA) [27].

### 2.9. Anthropometric Measurements

Anthropometric measures were taken at baseline, 8 weeks, and 16 weeks. Weight was measured to the nearest 10 g with an electronic scale (Tanita Scale, Tanita Corp., Arlington Heights, IL, USA, capacity 14 kg for infants and 36 kg adults, Tokyo, Japan) and length/height were measured to the nearest millimeter with a length board (Schorr Industries, Glenn Burney, MD, USA). Weight and length were transformed to Z-scores using the 2006 WHO reference standards [28]. Stunting was defined as a length for age Z-score < − 2 SD; underweight as weight for age Z-score < −2 SD; wasting as weight for length/height Z-score < −2 SD, and overweight or obesity as body mass index (BMI) for age Z-score > +2 SD [28].

### 2.10. Dietary Assessment

Dietary information was collected with a 7-day semi-quantitative food frequency questionnaire (SFFQ). Trained health personnel administered the SFFQ. In an interview, parents/caregivers were asked to recall the number of days of the week, times-a-day, and portion sizes that their children consumed over a 7-day period, based on standard and home-made measurements, as well as the number of portions consumed for each food item. Foods included in the questionnaire represented 95% of the total dietary consumption of pre-school-age children, according to a single 24 h recall questionnaire from the 1999 Mexican National Nutrition Survey (ENN-99) [29]. Thirty nine new foods were added to the Encuesta Nacional de Salud y Nutrición 2006 (ENSANUT-2006), some of which were adopted for use in the SFFQ. These foods and dishes were commonly consumed, and were classified according to fat, sugar and sodium content [24]. A group of experts in nutrition from the Center for Nutrition and Health Research in the National Institute of Public Health in Mexico selected the items used in the SFFQ, which included 123 foods and beverages. Consumption of foods was expressed in grams (g) or milliliters (mL) in one day. Consumption was calculated according to frequency, portion size (g or mL), and number of portions at each meal divided by seven days of the week. The following step was to convert the amount (g or mL) of each item into energy and nutrients using a food composition table, which was created by the National Institute of Public Health in Mexico [30].

### 2.11. Formula Intake

For each participant, total formula intake was divided by total days of study participation. The median of this individual average consumption was calculated and compared over the analysis sample for each study group. Bias-corrected bootstrap 95% confidence intervals for these statistics were obtained with 1000 replicates through stratified resampling by study group.

### 2.12. Sociodemographic Characteristics

Study population characteristics were obtained at baseline by interviewing the parents/caregivers. A socioeconomic index was constructed through principal component analysis using household conditions (floor and roof material), services and basic household infrastructure (e.g., sources and disposal of water availability of toilet, and gas stove) to create a score [31]. The first principal component explained 30% of total variance.

### 2.13. Morbidity

Parents were asked to provide information on their infant’s health on Monday, Wednesday and Friday. The question asked for diarrhea was: Did your infant have diarrhea yesterday? The question asked for respiratory tract infection was: Did your infant present cough, flu, or cold symptoms yesterday? If the infant presented more than three watery stools a day, he or she was sent to the nearest health center and the parent was asked to stop using the formula.

### 2.14. Sample Size

We approached the statistical power calculation by simulation with 1000 replicates by way of a median regression model and assuming a Gaussian distribution for the outcome in each study group. Given a total sample size of *n* = 180 (90 observations per group), this study achieved a statistical power of 80.7% to detect a median difference of 0.54 in standardized units.

### 2.15. Statistical Analysis

Baseline characteristics were compared between study groups through descriptive statistics. For continuous variables, the median, 25th and 75th percentiles were calculated. For categorical variables, frequencies and percentages were calculated for each category. Median changes in outcome variables and their difference between study groups were estimated through median regression using change in outcome as dependent variable, and an indicator variable of study group (0 = Non-PUFAs, 1 = PUFAS) as predictor. Median change for each study group was calculated with the appropriate linear combination of model coefficients and differences between study groups were calculated with the model coefficient of the study group indicator variable. Covariate-adjusted median change was obtained by adding the baseline measurement of the outcome and an indicator variable of sex (0 = male, 1 = female) as predictors. Covariate-adjusted median change for each study group was obtained with the appropriate linear combination of model coefficients by setting the baseline measurement predictor at its median and the sex indicator variable at its mean value for the whole sample, which corresponds to the proportion of female children. Standard errors were obtained by bootstrap resampling within each study group with 1000 replicates [32]. Bias-corrected 95% confidence bootstrap intervals were obtained for all estimates. Given the importance of the first 1000 days of life of a child, median regression models included an interaction between study group and age group to detect whether the effects on fatty acids varied between age groups (<24 months, 24–33 months); interactions of age group with the other predictors were also included. Stata^®^ IC 16.1 was used for all analyses (Stata Corp. 2019, Stata Statistical Software, College Station, TX, USA).

## 3. Results

The number of children assessed for eligibility were 271. A total of 232 healthy infants were randomized to receive the PUFAs-fortified or non-PUFAs-fortified formula. A total of 19 (16.1%) infants from PUFAs group and 20 (17.5%) in Non-PUFAs withdrew before the end of the study. The main reason for dropout was mothers refusing a second blood sample from their infant. In the group of 193 children who completed the study, 99 (51.3%) were assigned to the PUFAs group and 94 (48.7%) to the Non-PUFAs group. (Figure 1).

Both formulas were well accepted by the infants. The median average PUFAs intake over the days of observation was 440 mL (95% CI = 425, 462), and 421 mL (95% CI = 409, 448) for the Non-PUFAs formula group, with no significant difference between study groups (19 mL; 95% CI = −14, 44). The proportion of infants that reported diarrhea was 42.9% for PUFAs and 57.1% for Non-PUFAs (*p* = 0.498), and the proportion of infants with respiratory tract infection was 54.7% for PUFAs and 45.3% for Non-PUFAs (*p* = 0.384).

Baseline characteristics for each treatment group are presented in Table 2. In general, these characteristics were balanced between study groups except for sex; 60.6% were male in the PUFAs group whereas 47.9% were male in the Non-PUFAs group.

Table 3 shows median changes in percentage points (p.p.) of total whole-blood fatty acids (FA) from the serum lipid profile between baseline and after four months by treatment group, adjusted by sex and baseline measurement of the outcome. The PUFAs group showed significantly higher median change compared to the Non-PUFAs group for DHA (+0.29 p.p.), and Alpha-Linolenic acid (+0.06 p.p.).

Table 4 shows the median change in anthropometry and micronutrient outcomes for infants between baseline and after four months by treatment group, adjusted by sex and baseline measurement of the outcome. In both groups more than 50% of infants increased their length/height-for-age Z-score, with median change of 0.16 (95% CI = 0.07, 0.26) in the PUFAs group, and 0.23 (95% CI = 0.14, 0.33) for the Non-PUFAs group, and with no significant difference between treatment groups. As for micronutrients, median change of folate was significantly higher in the PUFAs group at −0.87 (95% CI = −1.38, −0.44), compared to the Non-PUFAs at −3.83 (95% CI = −4.65, −3.03), wherein a statistically significant difference of 2.96 (95% CI = 2.02, 3.84) was seen between the PUFAs and Non-PUFAs groups (Table 4).

Analysis of the variation of treatment effects on FA by age group (<24 months and 24–33 months) showed no statistically significant differences in covariate-adjusted models (*p* > 0.258), nor in models without adjustment covariates (*p* > 0.185).

## 4. Discussion

This double-blind, randomized controlled trial shows that infants consuming a milk-based formula containing PUFAs for four months significantly increased their percentage levels of DHA and alpha-linolenic acid compared with those who consumed Non-PUFAs-fortified formula. Anthropometrical outcomes showed that infants in both groups slightly increased in length/height-for-age Z-score, although the difference was not statistically significant. Micronutrient composition within each group differed in that folate levels were significantly higher in infants from the PUFAs group as compared to their counterparts.

These results reflect those reported in previous studies. In one double-blind, controlled randomized trial carried out in healthy term infants, researchers evaluated the impact of three different infant formulas containing a mix of dairy fat and plant oils, only plant oils, or plant oils supplemented with DHA and ARA (arachidonic acid from the omega-6 family) for a period of four months. The formula containing dairy lipids fortified with PUFAs significantly increased total serum membrane omega-3 levels [33]. Birch et al. conducted a double-masked, randomized trial with four infant formulas containing equal amounts of nutrients, but in different dosages of PUFAs: 0% DHA, or 0.32% DHA, 0.64% DHA and 0.96% DHA, over a 12-month period. Red blood cell (RBC) DHA concentrations were significantly different (<0.001) between all formula groups at both four and 12 months of age, and RBC DHA concentration increased as formula DHA dosage increased [34]. In an observational study which compared RBC membrane fatty acids in infants supplemented with DHA and ARA with other types of milk, results showed that infants consuming supplemented formula had significantly higher levels of DHA and other omega-3s, as well as lower levels of omega-6 fatty acids in RBC membranes, than infants consuming non-supplemented formula [35].

These studies have shown that DHA serum concentrations significantly increased in pre-school-age children who receive an infant formula supplemented with PUFAs [33,34,35]. Our results support the critical importance of PUFAs supplementation during infancy. One reason is that, in developing countries, there is a low consumption of foods rich in PUFAs, such as oil-rich fish [8]. Second, studies have shown that particularly during infancy, it is difficult to meet dietary PUFAs requirements [9,13]. Third, between the ages of six months and three years, infants may experience infantile anorexia nervosa, an eating disorder with onset during the early developmental stage of separation and individualization [36], wherein the infant refuses to eat in an attempt to achieve autonomy and control with regard to the mother [36].

Previous results of anthropometry and consumption of formulas rich in DHA are in accordance with our study. In Mexico, in a cohort study begun in 2006 and continuing to the present, pregnant women are given 400 mg/day of algal DHA and a soy and corn oil-based placebo for the last six months of pregnancy. Findings have shown no overall impact of DHA on infant weight and height at 18 months of age [37]. However, the study observed that among primigravid women who received DHA supplementation, infant length at 18 months did significantly increase 0.72 cm (95% CI = 0.11, 1.33), representing 0.26 length-for-age Z-score units [37]. The same study assessed prenatal supplementation with DHA on infant’s weight, length and body mass index through 60 months of age. Results showed no significant differences by treatment group for these anthropometric measurements at 60 months of age (all *p* > 0.05) [38]. On the other hand, a study carried out in the United States on healthy singleton term infants who received either infant formula supplemented with LCPUFAs or a placebo formula were re-enrolled at 18 months and given follow-up by anthropometric assessment until six years of age. Results at 18 months of life showed that infants fed LCPUFA-supplemented formula had significantly greater linear growth than their counterparts [34]. From two to six years of age, LCPUFA-supplemented infants had significantly greater height for age than their counterparts [39].

Results of growth after supplementation with PUFAs are not conclusive. Most studies justify the addition of PUFAs to infant formula citing vision and cognitive development [40]. Nonetheless, only length and weight have been reported as outcome indicators [40]. These studies have shown that supplementation of PUFAs during pregnancy, lactation and early infancy may improve birth weight, weight and length for infants in developing countries [40]. In children above two years of age, no benefits of PUFAs consumption through infant formula on growth were observed in studies from developed and developing countries [40]. There is little evidence on the effect on PUFAs supplementation in children above two years of age. There is a need to continue studying this age group [40].

Median change in folate serum concentration was significantly higher in the PUFAs infant formula group. This may be explained by high compliance levels during the trial and because the PUFAs infant formula had four times more folic acid than Non-PUFAs infant formula. PUFAs infant formula also contained vitamin B12, a co-factor which enhances folic acid status [41]. Therefore, folate serum concentration was significantly higher among the PUFAs formula group.

Strengths of our study include that it was a well-controlled supplementary feeding trial. There was high compliance to treatment, and milk-based formula was properly diluted according to guiding standards. Adherence to treatment was measured at the study site by daily logs and confirmed by the changes in lipid blood levels. The infant formula was also supplied free of charge to the participants, which encouraged adherence to the randomized dietary allocation. Anthropometry was measured with precision and accuracy by standardized personnel at all the day-care centers.

Dropout rate was lower than 20% during the four-month study period, and the proportion of these dropouts was balanced between groups. Each day-care center trained the personnel that supervised the amount of formula consumed by each child daily.

Our study had some limitations; the four months of follow-up could have been shorter. We did not measure mental or motor development in children. Finally, there was a lack of information on infants’ gestational age.

## 5. Conclusions

In this double-blind, randomized clinical trial, a significant improvement was observed in the lipid profile of children who received infant formula enriched with micronutrients and PUFAs, compared to children who received formula fortified with micronutrients only. In both groups, infants increased their length/height-for-age Z-score. Studies are needed on the effect of PUFAs-fortified infant formula on neurodevelopment, cognitive function, behavior and health outcomes (i.e., immune response, overweightness) of children.

## Figures and Tables

**Figure 1 nutrients-13-00004-f001:**
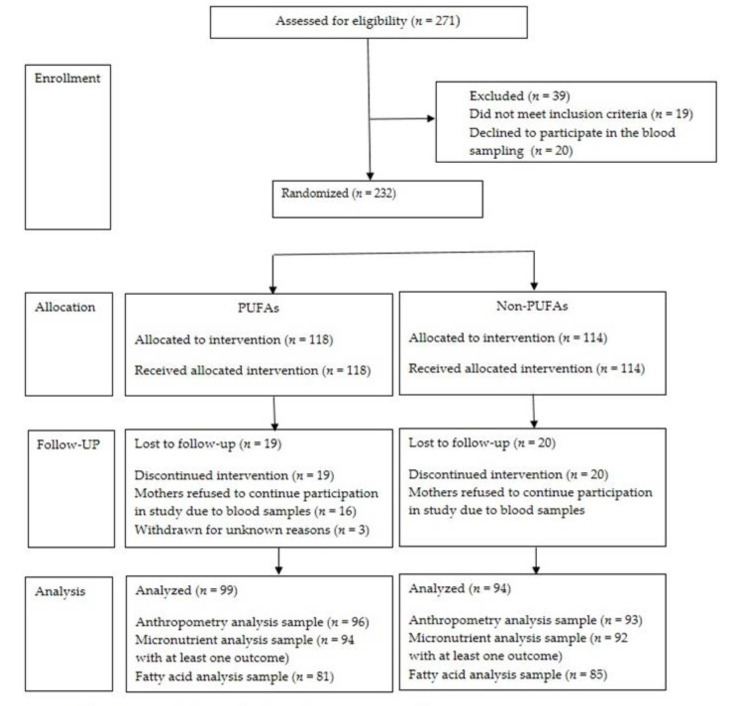
Flow of study subjects. Formula PUFAs contained vegetable oil (palm oil, coconut, soy and high oleic sunflower oils). DHA from fish origin, Linoleic Acid, Alpha-Linolenic Acid. Morbidity information included diarrhea and acute respiratory infection. PUFAs: Milk-based infant formula with polyunsaturated fatty acids, Non-PUFAs: Formula with micronutrients only non polyunsaturated fatty acids.

**Table 1 nutrients-13-00004-t001:** Nutrient composition of study formulas by treatment group.

Nutrients	Polyunsaturated Fatty Acids (PUFAs) (100 g Powder)	PUFAs Per Serving *	NON-PUFAs (100 g Powder)	NON-PUFAs Per Serving *
Energy (kj/kcal)	1894/447	757.6 /178.8	2193/524	877.2/209.6
Carbohydrates (g)	58.5	23.4	37.9	15.16
Prebiotics (Dietary fiber) (g)	3.0	1.2	---	---
Total Lipids (g)	16.6	6.64	29.6	11.84
Linoleic Acid (mg)	1550.0	620	---	---
Alpha-Linolenic Acid (mg)	195.0	78	---	---
DHA ^a^ (mg)	40.5	16.2	---	---
Protein (g)	17.9	7.16	26.7	10.68
Vit. A (µg retinol eq)	267.0	106.8	522.67	209.068
Vit. D3 (µg, cholecalciferol)	3.65	1.46	20.33	8.132
Vit. E (µg, tocopherol eq)	5.37	2.15	9.75	3.9
Vit. C (mg, ascorbic acid)	50.0	20	35.41	14.164
Vit. B1 (µg, thiamin)	763.0	305.2	370	148.0
Vit. B2 (µg, riboflavin)	1070.0	428	1500	600
Vit. B3 (µg, niacin)	5200.0	2080	720	288
Vit. B5 (µg, panthotenic acid)	3060.0	1224	3500	1400
Vit. B6 (µg, piridoxin)	494.0	197.6	220	88
Vit. B8 (µg, biotin)	19.50	7.8	20.1	8.04
Vit. B9 (µg, folic acid)	85.8	34.32	17.5	7.0
Vit. B12 (µg, cianocobalamin)	1.65	0.66	---	---
Vit. K (µg)	31.0	12.4	33.0	13.2
Choline (mg)	81.0	32.4	---	---
Taurine (mg)			---	25.6
Calcium (mg)	660.0	264.0	186.2	74.48
Phosphorus (mg)	600.0	240.0	690.0	276.00
Iron (mg)	6.9	2.76	0.3	0.12
Magnesium (mg)	57.5	23.0	74.0	29.60
Zinc (mg)	5.3	2.12	2.65	1.06
Iodine (µg)	125.0	50.0	1.36	0.544
Copper (µg)	360.0	144.0	---	---
Sodium (mg)	255.0	102.0	340.0	136.0
Potassium(mg)	020.0	408.0	1080.0	432.0
Selenium (µg)	10.5	4.2	---	---
Inositol (mg)	58.5	23.4	---	---

Mead Johnson^®^ 2010, México. Formula PUFAs contained vegetable oil (palm oil, coconut, soy and high oleic sunflower oils). DHA from fish origin, Linoleic Acid, Alpha-Linolenic Acid. * 240 mL water + 40 g powder. ^a^ Docosahexaenoic acid (22:6n-3). Vit.:vitamin, DHA: docosahexaenoic acid.

**Table 2 nutrients-13-00004-t002:** Baseline characteristics of infants who completed the study, by treatment group.

Variables		PUFAs (*n* = 99)		Non-PUFAS (*n* = 94)
	*N*	P50 (P25, P75) or %	*N*	P50 (P25, P75) or %
Age, months	96	21.27 (16.59, 27.88)	93	22.97 (16.53, 27.14)
<24	60	62.5	55	59.1
24–30	36	37.5	38	40.9
Sex				
Male	60	60.6	45	47.9
Female	39	39.4	49	52.1
Anthropometry				
Birth weight, kg	98	3.19 (2.70, 3.45)	89	3.10 (2.78, 3.50)
Weight, kg	96	11.20 (10.40,12.25)	93	11.50 (10.10, 12.80)
Height, cm	96	82.40 (78.15, 86.90)	93	83.70 (78.30, 87.60)
HAZ	96	−0.77 (−1.63, -0.04)	93	−0.58 (−1.58, 0.02)
Stunting ^a^	12	12.5	13	14
WAZ	96	−0.07 (−0.86, 0.55)	93	−0.04 (−0.63, 0.58)
Low weight ^b^	1	1	0	0
WHZ	96	0.35 (−0.34, 0.92)	93	0.34 (−0.20, 0.85)
Wasting ^c^	1	1	0	0
BMIZ	96	0.47 (−0.25, 1.08)	93	0.54 (−0.11, 1.00)
Overweight ^d^	4	4.2	7	7.5
Socioeconomic score	98	0.21 (−0.82, 0.89)	94	0.21 (−0.79, 0.89)
Biochemical indicators				
Hemoglobin, g/dL	96	12.2 (11.2, 13.0)	93	11.9 (11.0, 2.6)
<11	21	21.9	23	24.7
<10	10	10.4	9	9.7
Vitamin A, µg/dL	89	31.58 (26.83, 35.78)	88	32.12 (28.49, 35.25)
Vitamin D, nmol/L	92	37.60 (30.75, 58.44)	88	39.87 (31.06, 54.89)
Zinc, µg/dL	77	119.2 (111.7, 133.5)	71	120.6 (109.2,132.3)
Ferritin, µg/L	78	20.15 (12.85, 29.58)	72	17.99 (9.27, 24.86)
Folate, ng/mL	90	15.40 (12.60, 16.70)	89	14.90 (12.10, 16.70)
Fatty acids ^e^				
Estearic	81	7.74 (7.14, 8.37)	85	7.90 (7.24, 8.42)
Lauric	81	0.20 (0.11, 0.31)	85	0.25 (0.15, 0.31)
Linoleic	81	30.68 (28.29, 32.96)	85	28.19 (25.39, 30.52)
Alpha-Linolenic	81	0.55 (0.45, 0.65)	85	0.55 (0.49, 0.62)
Miristic	81	1.26 (0.95, 1.61)	85	1.57 (1.11, 2.19)
Oleic	81	25.64 (23.44, 27.45)	85	26.16 (24.63, 27.79)
Palmitic	81	26.65 (25.18, 28.99)	85	28.27 (26.34, 29.54)
Palmitoleic	81	2.03 (1.53, 2.52)	85	2.29 (1.82, 2.77)
EPA ^f^	81	0.26 (0.16, 0.32)	85	0.30 (0.22, 0.36)
DHA^g^	81	0.56 (0.42, 0.76)	85	0.59 (0.44, 0.77)
Dietary indicators				
Energy intake, kcal/day	95	1421.4 (901.49,1927.93)	92	1383.1 (993.99, 1719.59)
Carbohydrates, g/day	95	216.9 (150.3, 307.7)	92	217.5 (151.5, 286.8)
Proteins, g/day	95	43.2 (28.5, 60.5)	92	41.3 (27.9, 54.8)
Lipids, g/day	95	40.2 (23.2,57.2)	92	38.6 (23.7, 51.4)
Alpha-Linolenic (omega-3), g/day	95	0.31 (0.17, 0.48)	92	0.28 (0.18,0.43)
Gamma-Linolenic, g/day	95	0.02 (0.009, 0.02)	92	0.01 (0.008, 0.20)
EPA ^f^, g/day	95	0.007 (0.003, 0.018)	92	0.005 (0.002, 0.11)
DHA ^g^, g/day	95	0.02 (0.01, 0.05)	92	0.02 (0.005, 0.04)

^a^ HAZ (height-for-age) Z-score < −2 SD; ^b^ WAZ (weight-for-age) Z-score < −2 SD; ^c^ WHZ (weight-for-height) Z-score < −2 SD; ^d^ BMI Z-score > +2 SD overweight. ^e^ Results are reported as percentage of total whole-blood fatty acids (FA%), ^f^ Eicosapentaenoic acid (20:5n-3), ^g^ Docosahexaenoic acid (22:6n-3).

**Table 3 nutrients-13-00004-t003:** Covariate-adjusted ^a^ median change for FA% between baseline and after four months, by treatment group.

Outcome	PUFAs (*n* = 81)		Non-PUFAs (*n* = 85)		PUFAs vs. Non-PUFAs
	*N*	Median Change (95% CI)	*N*	Median Change (95% CI)	Difference (95% CI)
Arachidonic	81	−0.06 (−0.27, 0.14)	85	0.16 (−0.12, 0.66)	−0.22 (−0.69, 0.19)
Estearic	81	0.17 (−0.08, 0.41)	85	0.19 (−0.10, 0.38)	−0.02 (−0.36, 0.40)
Lauric	81	0.02 (−0.03, 0.05)	85	0.02 (−0.04, 0.04)	0.00 (0.00, 0.06)
Linoleic	81	−1.38 (−2.27, -0.43)	85	-0.94 (−1.72, 0.20)	−0.43 (−1.93, 0.78)
Alpha-Linolenic	81	0.08 (0.04, 0.12)	85	0.02 (−0.01, 0.04)	0.06 (0.02, 0.12)
Miristic	81	0.01 (−0.18, 0.19)	85	0.20 (0.00, 0.44)	−0.19 (−.50, 0.08)
Oleic	81	0.65 (−0.10, 1.41)	85	−0.54 (−1.27, 0.54)	1.18 (−0.35, 2.08)
Palmitic	81	−0.36 (−1.37, 0.29)	85	0.55 (−0.21, 1.19)	−0.91 (−2.16, 0.06)
Palmitoleic	81	0.08 (−0.13, 0.22)	85	0.14 (0.00, 0.28)	−0.06 (−0.36, 0.13)
EPA ^b^	81	0.05 (0.01, 0.08)	85	0.02 (−0.01, 0.04)	0.02 (−0.02, 0.07)
DHA ^c^	81	0.22 (0.17, 0.35)	85	−0.07 (−0.12, −0.03)	0.29 (0.22, 0.40)

Results are reported as median changes in percentage of total whole-blood fatty acids (p.p.); ^a^ Estimates obtained from median regression with sex and baseline measurements as adjustment covariates; Bootstrap bias-corrected 95% confidence intervals in parentheses. 1000 replicates were generated with stratified sampling by study group. ^b^ Eicosapentaenoic acid (20:5n-3). ^c^ Docosahexaenoic acid (22:6n-3).

**Table 4 nutrients-13-00004-t004:** Covariate-adjusted median changes ^a^ of anthropometric and micronutrient outcomes between baseline and after four months, by treatment group.

Outcome	PUFAs		Non-PUFAs		PUFAs vs. Non-PUFAs
	*N*	Median Change (95% CI)	*N*	Median Change (95% CI)	Difference (95% CI)
Anthropometry					
Height for age, Z	96	0.16 (0.08, 0.28)	93	0.23 (0.14, 0.33)	−0.07 (−0.19, 0.07)
Weight for age, Z	96	0.04 (-0.03, 0.12)	93	0.10 (−0.04, 0.19)	−0.05 (−0.16, 0.10)
Weight for height, Z	96	−0.03 (−0.14, 0.09)	93	0.00 (−0.14, 0.13)	−0.03 (−0.20, 0.16)
BMI for age, Z	96	−0.04 (−0.18, 0.08)	93	0.00 (−0.15, 0.15)	−0.04 (−0.25, 0.14)
Vitamins & minerals					
Vitamin A, µg/dL	89	1.19 (−0.77, 2.40)	88	−0.54 (−1.96, 0.84)	1.73 (−0.85, 3.77)
Vitamin D, nmol/L	92	6.25 −1.43, 16.02)	88	4.35 (0.08, 10.82)	1.90 (−8.88, 13.48)
Zinc, µg/dL	77	−2.42 (−6.36, 3.57)	71	−4.22 (−9.36, 1.44)	1.80 (−5.70, 10.17)
Ferritin, µg/L	78	−0.84 (−3.56, 3.27)	72	−4.84 (−8.32, 0.62)	4.00 (−3.11, 8.73)
Folate, ng/mL	90	−0.87 (−1.38, -0.44)	89	−3.83 (−4.65, −3.03)	2.96 (2.02, 3.84)

^a^ Estimates obtained from median regression with sex and baseline measurements as adjustment covariates. Bootstrap bias-corrected 95% confidence intervals in parentheses. 1000 replicates were generated with stratified sampling by study group.

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
