# Peer review of "Effect of Milk-Based Infant Formula Fortified with PUFAs on Lipid Profile, Growth and Micronutrient Status of Young Children: A Randomized Double-Blind Clinical Trial"

_nutrients, 2020, doi:10.3390/nu13010004_

Round 1
Reviewer 1 Report
Interesting work. However, the title must be changed to one that describes exactly the work done.
I have just one main doubt. It was not possible to modify just the PUFAs contente on both formulae? It was not possible to standardize energy, fat and protein levels per serving? Or there are specific reasons to use these specific formulations?
Some text editing is advisable and references should be carefully revised by the authors. Please find my other suggestions in the attached file.

Author Response
INSTITUTO NACIONAL DE SALUD PÚBLICA
December 11, 2020
Editorial Office
Nutrients
Manuscript ID: Nutrients-1011218
Title: “Effect of fortified milk on PUFAs, growth and micronutrient status in children 12 to 30 months old: A randomized, double-blind clinical trial”
The authors appreciate the reviewers’ valuable comments. We have incorporated all their suggestions.
Response to reviewers:
Reviewer 1.
Interesting work. However, the title must be changed to one that describes exactly the work done:
We appreciate this observation. The title has been modified as follows:
“Effect of milk-based infant formula fortified with PUFAs on lipid profile, growth and micronutrient status of young children: a randomized double-blind clinical trial”
I have just one main doubt. It was not possible to modify just the PUFAs content on both formulae? It was not possible to standardize energy, fat and protein levels per serving? Or there are specific reasons to use these specific formulations?
The nutrient compositions of both milk-based infant formulas meet the Mexican recommendations and standards. The Non-PUFAs supplement is a commercial product and we could not modify its content. Although this group did not receive DHA, they did received formula with micronutrients. In Mexico, stunting and micronutrient deficiencies are still prevalent mainly among vulnerable populations of low income, indigenous background, those living in rural areas, and those living in the southern region of the country. We used a formula with micronutrient content as a placebo because we wanted both treatment groups to receive a benefit.
Some text editing is advisable and references should be carefully revised by the authors. Please find my other suggestions in the attached file.
Edits and references have been carefully reviewed by a native English speaker.
Abstract:
- - PUFAs should be defined the first time written:
We appreciate this observation. In response, the word “PUFAs” has been defined. Please see changes. Line: 19
- - Milk 240mL accordingly:
We appreciate this observation. In response, the number (240mL) and the wording has been changed as suggested. Line: 24-25
“Children received prepared formula (240mL) twice a day, according to the color-code assigned to each infant”
- - Infants increased….for age Z-score. Median change…..
We appreciate this observation. In response, see Lines 31-32.
“Infants increased their length/height-for-age Z-score: median change for the PUFAs group was 0.16 (95%CI =0.08, 0.28) and 0.23 (95%CI= 0.14, 0.33) for Non-PUFAs, with no differences between groups”.
Introduction:
- - Do you really mean performed?
We appreciate this observation. The word performed was spelled incorrectly. In response, the correct word is preformed. see Line: 51.
- For Lines 52 & 57 punctuation have been changed.
are transferred to the fetus and infant by the placenta and/or mammary gland. Low levels of PUFAs may compromise maternal stores [7]. According to the National Health and Nutritional Survey-2006, more than 50% of the population (all ages) does not meet the recommended dietary intake of PUFAs. Median consumption of DHA and EPA were 30 mg/d [8].
- - Lines 78 & 80 have been modified from “Moreover, anthropometry and micronutrient status as a secondary endpoint” to:
The main objective of this study is to assess the effect of PUFA-fortified milk-based infant formula on lipid levels in Mexican infants 12 to 30 months old. Moreover, anthropometry and micronutrient status of the infants were evaluated.
Line: 78
Material and Methods:
1.-Table1. Align text with other column.
Text has been align with column.
2.- (<99%) ?
Thank you for your observation. The number expression “(<99%)” has been revised to read:
Line: 158 purity (>99%) standard mixtures (Sigma-Aldrich Chemie GmbH, 37 FAs mixture).
- - Thank you for your comment. The (TOSOH) and (CMIA) have been defined in lines 165-167 as follows:
“measured by Toyo Soda Manufacturing Company (TOSOH ) automated immunoanalyzer. Vitamin D: 25-OH-D3 was analyzed by chemiluminescent microparticle immunoassay (CMIA)”.
- - Thank you for your comment. The (SFFQ) has been defined in Line 180-181 as follows:
Semi-Quantitative Food Frequency Questionnaire (SFFQ).
- - Thank you for your comment. RBC has been defined on Line 293 as follows:
Red Blood Cells (RBC)
- - Thank you for your comment. LPUFAs has been defined on Line 320-323 as follows:
Long-Chain Polyunsaturated Fatty Acids (LCPUFAs).
- - Thank you for your comment. The sentence has been corrected in lines 331 & 333w as follows:
There is little evidence on the effect on PUFAs supplementation in children above 2 years of age. There is a need to continue studying this age group.
Reviewer 2 Report
This manuscript presents the results of a randomized double-blind clinical trial where children 12 to 30 months old were randomized to receive 240 mls/day of milk infant formula with or without PUFAs. A big, big study for such a little result.
Page 5 lines 171-172: the difference betwen underweight and wasting is not clear
Author Response
Reviewer 2.
This manuscript presents the results of a randomized double-blind clinical trial where children 12 to 30 months old were randomized to receive 240 mls/day of milk infant formula with or without PUFAs. A big, big study for such a little result.
- - Page 5 lines 174-177: the difference between underweight and wasting is not clear
Thank you for your comment. The difference between underweight and wasting has been specified as follows: Lines 174-177
Stunting was defined as a length for age z-score <- 2 S.D.; underweight as weight for age z-score <-2 S.D.; wasting as weight for length/height z-score <-2 S.D., and overweight or obesity as body mass index (BMI) for age z-score >+ 2 S.D.[23].
Reviewer 3 Report
The manuscript entitled "Effect of fortified milk on PUFAs, growth and micronutrient status in children 12 to 30 months old: A randomized, double-blind clinical trial" is a good study that will contribute to the current literature.
- Lines 158-165, “2.8. Micronutrients”: Here the methods that are used to measure contents of zinc, folate, vitamin D and vitamin A needs to cited. In addition, please also indicate the units of expression for the micronutrients.
- In the discussion, some results were not expressed based on statistical evaluation. E.g., line: “higher level”, line 275: “higher”. Are these changes statistically significant? Please go through the entire manuscript and makes sure that all results are discussed based on statistical evaluation.
- The conclusion section can be extended. Please make recommendations for future research. What more needs to be done?
- In addition to the above, the language of the manuscript needs to be improved. E.g., lines 20-21: Incomplete sentence; lines 43-44, 48-50: Please re-write these sentences.
Author Response
INSTITUTO NACIONAL DE SALUD PÚBLICA
December 11, 2020
Editorial Office
Nutrients
Manuscript ID: Nutrients-1011218
Title: “Effect of fortified milk on PUFAs, growth and micronutrient status in children 12 to 30 months old: A randomized, double-blind clinical trial”
The authors appreciate the reviewers’ valuable comments. We have incorporated all their suggestions.
Response to reviewers:
Reviewer 3.
The manuscript entitled "Effect of fortified milk on PUFAs, growth and micronutrient status in children 12 to 30 months old: A randomized, double-blind clinical trial" is a good study that will contribute to the current literature.
Material and Methods:
- - Lines 158-165, “2.8. Micronutrients”: Here the methods that are used to measure contents of zinc, folate, vitamin D and vitamin A needs to cited. In addition, please also indicate the units of expression for the micronutrients.
Thank you for your comment. Please see lines 161-169. References have been cited and units of expression have been included.
“Serum concentrations of ferritin (µg/L) were measured by an immunoassay method using commercial kits (Dade Behring Inc., Deerfield, IL, USA), while concentrations of zinc (µg/dL) were determined by atomic absorption spectrometry using an Analyst 300 spectrometer (Perkin-Elmer, Norwalk, Ct, USA)[23]. Folate (ng/mL) was transformed into cyanocobalamin and stable folates. Concentrations were measured by Toyo Soda Manufacturing Company (TOSOH) automated immunoanalyzer [24]. Vitamin D measured through 25-OH-D3 (nm/L) was analyzed by chemiluminescent microparticle immunoassay (CMIA) using an Abbott Architect module [25,26]. Vitamin A (µg/dL) determinations were performed by high-performance liquid chromatography (HPLC) in a Waters instrument (Waters Co., Milford, MA, USA) [27]”
Discussion:
- In the discussion, some results were not expressed based on statistical evaluation. E.g., line: “higher level”, line 275: “higher”. Are these changes statistically significant? Please go through the entire manuscript and makes sure that all results are discussed based on statistical evaluation.
Thank you for your comment. We have reviewed the entire manuscript and the studies cited in this paper, and the word “significantly” or “statistical significance” was added when results showed statistical significance.
Conclusion:
- The conclusion section can be extended. Please make recommendations for future research. What more needs to be done?
Thank you for your comment. Please see lines 353-358 for future research. A brief statement about future research needs has been added at the end of the conclusion.
“In this double-blind, randomized clinical trial, a significant improvement was observed in the lipid profile of children who received infant formula enriched with micronutrients and PUFAs, compared to children who received formula fortified with micronutrients only. In both groups, infants increased their length/height-for-age Z-score. Studies on the effect on neurodevelopment, cognitive function, behavior and health-related outcomes (i.e. immune response, overweightness) of children who receive PUFAs-fortified infant formulas are needed.
Editing:
- - In addition to the above, the language of the manuscript needs to be improved. E.g., lines 20-21: Incomplete sentence; lines 43-44, 48-50: Please re-write these sentences.
Thank you for your comment. We have edited the manuscript correcting for language(up-load file). A native English speaker has reviewed the document.